# Questions of Identity in Sport Psychology Scholar–Practitioners

**DOI:** 10.3390/sports11090182

**Published:** 2023-09-13

**Authors:** Karen Howells

**Affiliations:** Cardiff School of Sport and Health Sciences, Cardiff Metropolitan University, Cardiff CF23 6XD, UK; klhowells@cardiffmet.ac.uk; Tel.: +44-29-2041-7153

**Keywords:** applied practice, BASES, BPS, HCPC, pracademic, sport psychologist

## Abstract

As with other academic disciplines, sport psychology academics working in higher education (HE) in the United Kingdom (UK) in lecturer and senior lecturer positions are typically required to hold a PhD in sport psychology or a related discipline. To work in applied practice with athletes, coaches, National Governing Bodies (NGBs), and sporting organisations, practitioners are required to acquire a qualification that affords registration with the Health and Care Professions Council (HCPC) through either the British Psychology Society (BPS) or the British Association of Sport and Exercise Sciences (BASES). Accordingly, scholar–practitioners, who have “a foot in both worlds” (Tenkasi and Hay, 2008), are required to have two related but distinct qualifications, each of which requires considerable resources (i.e., time, finances, and commitment) to achieve. This paper addresses some of the dilemmas and conflicts that these individuals may encounter in their primary workplace, which typically does not provide for applied practice (either in time or financial incentives). Specifically, issues around the knowledge-transfer gap will be addressed. Real-world examples will be in the form of reflections from the author’s own experiences. I am a senior lecturer in sport and exercise psychology at Cardiff Metropolitan University and the programme director of the MSc Sport Psychology. The role requires me to be HCPC registered, as well as have a PhD in sport psychology. I am also an HCPC Practitioner Psychologist, registered following completion of the BPS Qualification in Sport and Exercise Psychology (QSEP). My practice is limited to minimal private work and the supervision of trainee sport psychologists (BPS). At the end of the paper, I leave the reader with three questions to prompt reflection on what being a sport psychologist means and what contributions scholar–practitioners may offer to academic institutions and the clients we work with.

## 1. Questions of Identity in Sport Psychology Scholar–Practitioners

Rewind a decade—I was a mature student undertaking both my PhD and my professional qualification in sport psychology at a well-renowned academic institution in the United Kingdom (UK). Sometime into my learning, I was asked by my supervisor, who was unusually supervising both my PhD and professional qualifications, what I wanted to be known as—a researcher/educator or a practitioner with an emphasis on the *or*. The question perplexed me as it implied I could not be both, despite my engagement in and commitment to both academic and professional qualifications. I reflected, potentially frustrating my supervisor, that I wanted to be both. Now, a decade on, my personal tensions between the roles, on the one hand of scholar/academic, and practitioner on the other, remain, and in some respects, have become more conspicuous. These tensions go beyond the financial—paying my mortgage—or my contractual commitments. In the first instance, I am a full-time academic (with teaching and research responsibilities) at a post-1992 university; in the United Kingdom, a post-1992 university is synonymous with new university, it is often a former polytechnic or academic institution that was given university status through the Further and Higher Education Act 1992. Additionally, I work as an independent sole trader within the sport industry as a practitioner psychologist (that is, a Chartered British Psychological Society (BPS) Sport Psychologist). These tensions raise questions around identity—who am I? What do I represent? What is my value? In this paper, through reference to the wider literature on scholar–practitioners, through engagement with the legal/regulatory positionings of the Professional, Statutory, and Regulatory Bodies (PSRBs), and through personal reflection, I will address the existing requirements to be a scholar and a practitioner in the field of sport psychology in the UK. I will also consider the nomenclature associated with sport psychology identity and then address how those, like myself, who have “a foot in both worlds” [1]) aspire to research, teach, and practice. In doing so, I hope to negotiate the questions of identity that arise in inhabiting that interface between knowledge and practice. Finally, I will close with some questions to stimulate discussion about how scholar–practitioners can thrive at an individual level, but also contribute added value to both sport psychology education and applied practice.

## 2. Becoming a Sport Psychologist

The sport psychology discipline in the UK comprises individuals who inhabit many different physical and metaphorical spaces; we work in education, in or for National Governing Bodies (NGBs), in sports clubs, and in business. We may identify as educators, lecturers, psychologists, practitioners, or ascribe to a multitude of other labels that carry with them values, expectations, and indicators of our philosophies and motivations. In the UK, we are represented by a PSRB, the Division of Sport and Exercise Psychology (DSEP), a division of the BPS that promotes the professional interests of sport and exercise psychologists and aims to support the development of psychology both as a profession and as a body of knowledge and skills. A variety of individuals fall under this grouping, including, but not restricted to, Health and Care Professions Council (HCPC) registered practitioner psychologists (sport and exercise), university lecturers, and academic researchers. In this psychological melting pot of roles, responsibilities, and interests, what we identify with and how we present that identity impacts not only how we are viewed, but also the career opportunities available to us. So, in addressing issues of identity, nomenclature matters. 

Sport psychology, once mostly confined to the delivery of a plethora of mental skills promising to facilitate the “marginal gains” required for high performance in sport, has become a discipline that is concerned with both high performance and the wellbeing of those involved in sport, whether they are athletes, coaches, or practitioners. Like many students in sport science disciplines, it is typical for aspiring sport psychologists or academics to enrol in a postgraduate programme (e.g., MSc Sport and Exercise Psychology; MSc Sport Psychology; MSc Applied Sport Psychology) following successful undergraduate study in a sport or psychology-focused degree. Master’s programmes in the UK are typically accredited by the British Psychological Society (BPS) as Stage 1 of a two-stage route to becoming a sport psychologist [2]. Specific requirements of these programmes include preparing the students for the second stage (Stage 2), which comprises supervised practice for neophyte trainee sport psychologists. Accordingly, many MSc programmes will have modules that explicitly focus on associated learning outcomes. For example, the MSc Sport Psychology programme at Cardiff Metropolitan University (see Box 1) includes modules named Theory to Practice in Sport Psychology, Professional Development and Practice in Sport Psychology, and/or Counselling Approaches and Skills for Psychology Consultancy. These modules specifically address theory and content that lay the foundations for applied practice. 

Box 1Cardiff Metropolitan University: A case.Cardiff Metropolitan University, MSc Sport Psychology—A CaseCardiff Metropolitan University is a highly respected post-1992 university in respect of the delivery of sport programmes. It has a world recognised MSc Sport Psychology Programme which is accredited by the BPS as a Stage 1 programme. Although it, like other institutions in the UK has its own unique selling points, conforming to the stringent requirements of accreditation, it offers modules including Theory to Practice, Professional Development and Practice, and Counselling. Like all other BPS sport (and exercise) psychology accredited programmes there is a requirement that the Programme Director is a HCPC registered practitioner in addition to being a practicing academic. As a ranked REF institution, it prides itself on the production of 3- and 4-star research papers and staff can advance along the research career progression route—Lecturer, Senior Lecturer, Reader, Professor, although some choose an L&T career route aspiring to a Principal Lecturer position as opposed to Reader. Students on the MSc programmes are taught by academics, the significant majority of whom have doctorates, who are world leading researchers in their respective fields, a major strength of the programme. Accordingly, most of the staff, like at other higher education institutions with university status, have doctorates in their respective areas.

These practice-focused modules (and others) are designed and delivered by staff who are appointed as lecturers, senior lecturers, readers (principal lecturers), or professors—that is, they are scholars—and have sufficient knowledge and expertise in the content delivered in the modules. In line with other academic disciplines in higher education (HE) in the UK, these sport psychology academics, who are expected to deliver on postgraduate taught (PG-T) programmes, are typically required to hold a PhD in sport psychology or a related discipline (e.g., motor skills). The exception to this relates to the Programme Director (PD) role on a BPS-accredited MSc Sport and Exercise Psychology programme who is also required to be an HCPC-accredited practitioner psychologist. Therefore, it is necessary for the PD to hold both a PhD and a professional qualification; inter alia, they are required to be both a scholar and a practitioner, possessing two related but distinct qualifications, each of which requires considerable resources (i.e., time, finances, and commitment) to achieve. Despite the requirement for the PD to be a practitioner, the contractual obligations of academic institutions are often restricted to research and learning and teaching (L&T). Any applied practice that they engage in is typically performed outside the institution, either on short-term or private contracts. Beyond direct applied practice, it may involve assessing or supervisory responsibilities for the BPS or BASES; however, these activities are expected to be in addition to the primary responsibilities of the academic. There are exceptions to this, for example, when Talented Athlete Scholarship Scheme (TASS) athletes are supported by full-time members of HCPC-accredited academic staff at some sport-focused universities. Additionally, there is a trend towards increased opportunities for applied work under the guise of community engagement, although these are few and far between. So, it is clear that a minority of individuals in academic positions are required to be both a *scholar* and a *practitioner* even though the practitioners’ applied work does not typically fall under the remit of the academic institutions.

## 3. Scholar vs. Practitioner—What’s in a Name?

*Scholars* are those professionals who occupy and work in the geographical space of an academic institution and engage in research, teaching, and/or learning. Often, they will be academics inhabiting roles of educators, academics, lecturers, or researchers and possess (or be working towards) higher-level qualifications such as PhDs. Their pursuits, particularly when related to research, are primarily focused on knowledge acquisition and knowledge production. *Practitioners* are those who use their expert knowledge in a specified space to practice with clients, in the case of sport psychology practitioners, with national governing bodies (NGBs), coaches, athletes, or parents. To practice within the legal framework in the United Kingdom, sport psychology practitioners (or consultants) typically will have, or will be working towards, professional qualifications such as the BPS Qualification in Sport and Exercise Psychology (QSEP) or the BASES Sport and Exercise Accreditation Route (SEPAR). These qualifications enable the practitioners to apply for HCPC registration, bestowing on them the legal right to practice. In their consultancy work, they apply psychology in a variety of sport settings and in the field of motor skill research performance [3].

In the sport psychology domain, a gulf exists between scholars and practitioners, as Friesen [4] reports that despite “an implied direct connection between the scholarly literature and applied practice, [there] . . . [is a lack of] an empirical account of what practitioners believe to have been the most impactful scholarly writings to their applied practice” (p. 250). This has been acknowledged in the sport performance literature, with Holt et al. [5] identifying a gap between knowledge and practice and a culture whereby practices are not informed by the research. This is consistent with the wider research, for example, in marketing, where Harrigan and Hulbert [6] concluded that there was a disconnect between marketing education and marketing practice. To address this gulf, they argued that there was an impetus for marketing education to respond to the needs of its stakeholders, essentially marketing practitioners. For sport psychologists, adopting a stakeholder perspective such as this requires us to reflect on the objectives of institutions and programmes in preparing students for employment. It also encourages researchers to consider how their research impacts those who may utilise the findings and subsequent recommendations in their practice. One may argue that this process has already been implemented in the *theory to practice* focus of MSc Sport Psychology modules; however, the individuals designing and delivering these modules are not required and often do not have any applied practice experience themselves.

Postgraduate students on MSc programmes interested in a career in sport psychology must, at around the point of graduation from an MSc course, decide about which route to follow—a PhD, a professional qualification, or an alternative career path. Few have the financial or time resources to complete both a PhD and a professional qualification at the same time. I was fortunate that my personal circumstances at the time enabled me to pursue both. However, for the majority, the binary conceptualisation is perpetuated from the offset, and most, although there is no research to the best of my knowledge from the UK that has identified the qualifications and numbers of those in academia vs. practice, most choose one or the other route. Some who aspire to both engage in scholarly endeavours and applied practice manage to acquire both qualifications and divide their time between academia and applied practice, managing part-time academic positions alongside part-time consultancy contracts. However, they appear, anecdotally, to dedicate their time to one or the other, as, particularly in academic circles, a part-time contract that would enable time to practice may reduce the potential for academic career progression. Therefore, this may leave those like me who pursue both, albeit focusing on one endeavour more than the other, sometimes uncertain of where we fit yet bound by our values, ambitions, and interests to pursue both paths. Bouck [7] discussed this in relation to liminality, stating:

Scholar–practitioners evolve within a state of liminality characterized by feelings and emotions that range from ambiguity, questioning, confusion, and apprehension to openness, understanding, acceptance, and intentional disruption of the status quo. This process, while often unnerving, is critical to ultimate growth and proves productive if guided by the tenets of scholarly practice (p. 203).

Where this leaves the profession is that even for those, like myself, who wish to inhabit both roles, to do so is challenging and, in most cases, results in individuals defaulting to one ‘camp’ or the other. Yet, intuitively, and from the evidence, there is a strong argument for individuals to experience and inhabit both spaces as a scholar–practitioner [7] a scientist–practitioner [8], or, using the term more familiar to sport psychologists, a pracademic [9]. In the wider literature, there has been considerable discussion about the role of scholar–practitioners as bridging the gap between research and practice [10] or operating reflexively in the boundaries between theory and practice [11]. Nevertheless, there is an inherent tension that the scholar–practitioner is neither wholly scholar, who is motivated by a desire to produce knowledge, understand, and explain, nor wholly practitioner, who is driven by the desire to help, improve, and enhance the experiences of the individuals they are consulting with. Rather, Bailey [11] refers to what scholar–practitioners do as being at “the hyphen that joins the two words, where the two aspects of the same individual conjoin, where actions are guided by theory and theory is tempered by actions” (p. 50). Rather, adopting a pragmatic stance, they both make meaning and apply it to practice, seeking to bring about change, social justice, and challenge the status quo, a “sort of intellectual handyman” (p. 41). Effective scholar–practitioners must possess theoretical knowledge to be able to apply evidence-based practice, but they must also have sufficient understanding of the principles, culture, and social milieu of the organisations in which they operate. Furthermore, as Green [12] intimates, practitioners must be able to deliberate the external validity of published research and address whether it is practicable to apply it to specific situations. Collins and Collins [9] offer the view that the pracademic can actively bridge what they term the “research gap” (p. 5) by offering stakeholders interdisciplinary knowledge that is both evidence-informed and cognisant of the context (e.g., coaching). Nevertheless, this assumes that practitioners with access to extant and contemporary research will automatically apply the evidence to their practice solely by virtue of working with stakeholders in the field. For the sport psychologist pracademic/scholar–practitioner working in the field of sport and performance, one must possess both a high level of academic knowledge, for example, about resilience, emotional regulation, or choking, usually represented by the award of a doctorate-level qualification, as well as knowledge of the specific context and environment and the skills required to deliver in practice. Furthermore, this dual engagement requires them to be aware of both procedural ethics (e.g., specific institutional requirements; BPS practice guidelines) and ethics-in-action (i.e., the application of ethics in practice).

However, the reality is perhaps somewhat different. For scholar–practitioners who pursue a career in academia, it is reasonable to expect that their knowledge (albeit potentially within a restricted research interest area) increases, but with this is a risk of skill fade in applied practice as time and opportunity restrict their practice to being supervisors or assessors on the professional qualification route(s). This has wider implications; it means that in some instances, those academics who are qualified as supervisors and are named on the Register of Applied Psychology Practice Supervisors (RAPPS) may have little recent applied practice to inform their supervision; this may be conversationally considered the *blind leading the blind.* This concern led Wagstaff and Hays [2] to state the following:

Supervision remains up to speed with the contemporary demands of applied practice, we recommend ongoing supervisor quality assurance and training. Supervision and assessment must keep in touch with [the changing landscape] to stay rigorous, relevant, and respected over the next 10 years (p. 36).

For those who pursue a career in practice, their applied skills and experience increase, but their engagement with academic literature and, therefore, contemporary knowledge potentially wanes. To illustrate, most practitioners have worked with injured athletes at some point in their applied practice. However, in this area, Everard et al. [13] identified a knowledge-transfer gap between research and practice. They noted that research is not reaching end-users (i.e., athletes, coaches, and practitioners). Specifically, they identified that published research behind paywalls limits access to research findings. Accessing contemporary research is often cost-prohibitive for individual practitioners on more than an occasional basis—a 48-h access to one PDF article typically costs around GDP 37.00, and 30-day access to a full issue costs GDP 132.00 (e.g., Journal of Applied Sport Psychology). Academics, conversely, have almost unlimited access through their libraries and learning centres to a wealth of academic peer-reviewed journals. Where practitioners aspire to bridge the gap and engage in research or submit case studies for publication, their research is often viewed as lesser than the research findings submitted by academics [9]). This is particularly evident in respect of the publication of applied papers [14,15] in practitioner-focused journals such as *Case Studies in Sport and Exercise Psychology* (*CSSEP*), which “is a journal focused on providing practitioners, scholars, students, and instructors with case studies demonstrating different approaches and methods relevant to applied sport and exercise psychology” (CCSEP). Papers published in journals such as these, despite being peer-reviewed and providing useful, practical, and evidence-based information for neophytes and experienced practitioners alike, are not normally considered sufficiently robust or rigorous to be entered into the Research Excellence Framework (REF) submissions in the UK. The REF is undertaken by four funding bodies in the UK to secure the continuation of a world-class, dynamic and responsive research base across the full academic spectrum. Importantly for academic institutions the results inform the selective allocation of funding. Three elements are assessed: the quality of outputs, their impact beyond academia, and the environment that supports research. Thus, this creates an added dilemma for the scholar–practitioner as the REF requirements and demands of the scholar are often at odds with the practical research dissemination by the practitioner to interested stakeholders such as NGBs.

## 4. A Question of Identity

This paper so far has addressed the qualifications required by sport psychologists and discussed what we do and the decisions we must make about *what* roles we choose to inhabit. Yet the reality is more personal and involves questions of identity. Who are we? Scholar? Practitioner? Scholar–Practitioner? Quartiroli et al. [16] examined what they considered the under-researched construct of professional identity within sport psychology and lamented that a failure to address this posed a risk to the future of the profession. They described professional identity as an understanding and integration of generally agreed-upon professional philosophies and a scope of practice that is consistent with the consultant’s personal values and beliefs, which have evolved over the course of a career. Accordingly, when addressing the identity relating to scholar–practitioner, the answers, I argue, lie less in what we do and more in our informing beliefs and values. Bouck [7] explains, “in investigating the development of one’s identity as scholar–practitioner it is necessary to first come to an understanding of those elements that have shaped individual identity as well as the meaning of this identity as it relates to self and others” (p. 208). Reflection on philosophy and informing paradigms at the postgraduate level (Stage 1 and Stage 2) may begin to address the inherent tensions. In his discussion around liminality in scholar–practitioners, Bouck explained how exploring his beliefs and values in both his personal life and those related to education converged to a humanistic positioning that allowed him to “be more authentic and transparent in all aspects of my life” (p. 205). 

This resonates with my own experiences. When conducting my research, I identify as a social constructivist, interested in and valuing participants’ constructions of their own realities. Social constructivism is concerned with how individuals use language to generate their accounts of reality. Yet as Raskin [17] argues, they do not do this in isolation; “they do it together” (p. 121). My research utilising autobiographies [18,19], Olympic swimmers telling their stories, and foregrounding participants’ voices in focus group research exploring Olympians’ experiences of the post-Olympic Blues [20] reflects this philosophy. In my applied work [14], I have embraced a humanistic approach to my practice, adopting an athlete-centred approach in which I aspire to present an authentic, empathetic, and non-judgemental positioning whereby I encourage and support client autonomy [21]. As a practitioner, beyond my supervisory and assessing commitments, reflecting my own sporting interests in swimming, triathlon, and race sports, I tend to work with individual clients from individual sports rather than with teams. However, as humanistic psychologists have long argued, I acknowledge that humans have evolved as fundamentally cooperative and prosocial beings (Raskin, 2012). Therefore, it is important to me that in my practice I address and focus upon the impact of others’ influences in understanding human experiences and the motivations that my clients have to engage with coaches, teammates, and family. Therefore, in both research and practice, with a focus on the individual in the context of their wider social interactions, there is little doubt for me that these philosophies are complementary. 

## 5. Final Thoughts

In this paper, I have addressed the issues that I and others who identify as scholar–practitioners may have encountered with our feet in both the worlds of research (and learning and teaching) and practice. I hope that in addressing them, I have given the reader some ‘food for thought’. As this paper draws to a close, I leave you with some questions, informed by the discussion to date, that could prompt reflection on what being a sport psychologist means and what contributions scholar–practitioners may offer to academic institutions and the clients we work with.

The term scholar–practitioner has not yet entered common parlance; could discussing the option of this dual positioning, especially in relation to philosophy and informing paradigms at the postgraduate level, start to bridge the knowledge-transfer gap?Addressing the knowledge-transfer gap is critical to ensure that practitioners who operate outside academic institutions can access extant and contemporary research in relevant areas. How can the regulatory bodies, specifically BPS and BASES, support practitioners further to ensure that they have access to the research required to support effective practice?We could argue that HCPC accreditation with recent and relevant applied practice is more appropriate than a PhD for the delivery of some aspects of a post-graduate programme. A higher research degree may suggest an in-depth understanding and knowledge of a very narrow and restricted topic area, which may limit its relevance to applied practice. Therefore, should applied and practitioner-focused modules and learning materials on MSc programmes, that are accredited as the first stage to provide the building blocks for practice be designed and delivered by those professionals who have knowledge and expertise of recent and contemporary practice?

## Data Availability

Not applicable.

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
