# Peer review of "Questions of Identity in Sport Psychology Scholar–Practitioners"

_sports, 2023, doi:10.3390/sports11090182_

Round 1

Reviewer 2 Report

The author, predominantly via an autobiographical account, raises important questions of identity among sport psychology scholar-practitioners. Although her paper may not be of widespread interest among the global sport psychology community, it will be of great interest to those following a similar career path in the UK.

Although the quality of English is generally good, there are some areas in which it should be improved.

1. Be consistent in your use of hyphens. For example, you refer to Scholar Practitioners in the title and scholar-practitioners elsewhere. Similarly, check compound adjectives (e.g., peer-reviewed journals) throughout.

2. Be consistent in your use of capitals for organisations (e.g., national governing bodies).

3. You only need to write acronyms in full on the first occasion (e.g., UK, NGB).

4. I'm unsure why the font size varies throughout the paper.

5. Check the congruence between the in-text citations and the reference list (e.g., BPS, 1997 is not in the reference list).

6. Sentence structure is awkward in places (e.g., page 7, lines 30-34) and should be revisited.

7. Some phraseology is inelegant (e.g., page 6, lines 22-23 using discussed twice in the same sentence) and should be revisited.

8. On page 7, line 14 the author refers to her paper as this "chapter" causing me to wonder whether the paper is, in fact, a reworked chapter by the same author.

Specific errors:

1. Page 2, line 42 should be practice not practiced.

2. Page 4, line 20 add a comma after practice.

3. Page 7, line 21 remover space after - in knowledge-transfer.

Although the quality of English is generally good, there are some areas in which it should be improved.

1. Be consistent in your use of hyphens. For example, you refer to Scholar Practitioners in the title and scholar-practitioners elsewhere. Similarly, check compound adjectives (e.g., peer-reviewed journals) throughout.

2. Be consistent in your use of capitals for organisations (e.g., national governing bodies).

3. You only need to write acronyms in full on the first occasion (e.g., UK, NGB).

4. I'm unsure why the font size varies throughout the paper.

5. Check the congruence between the in-text citations and the reference list (e.g., BPS, 1997 is not in the reference list).

6. Sentence structure is awkward in places (e.g., page 7, lines 30-34) and should be revisited.

7. Some phraseology is inelegant (e.g., page 6, lines 22-23 using discussed twice in the same sentence) and should be revisited.

8. On page 7, line 14 the author refers to her paper as this "chapter" causing me to wonder whether the paper is, in fact, a reworked chapter by the same author.

Specific errors:

1. Page 2, line 42 should be practice not practiced.

2. Page 4, line 20 add a comma after practice.

3. Page 7, line 21 remover space after - in knowledge-transfer.
